# Revisiting Born’s Rule through Uhlhorn’s and Gleason’s Theorems

**DOI:** 10.3390/e24020199

**Published:** 2022-01-28

**Authors:** Alexia Auffèves, Philippe Grangier

**Affiliations:** 1Institut Néel, 25 rue des Martyrs, BP166, CEDEX 9, F38042 Grenoble, France; alexia.auffeves@neel.cnrs.fr; 2Laboratoire Charles Fabry, Institut d’Optique Graduate School, Centre National de la Recherche Scientifique (CNRS), Université Paris Saclay, F91127 Palaiseau, France

**Keywords:** quantum mechanics, contextuality, Gleason’s theorem, Uhlhorn’s theorem

## Abstract

In a previous article we presented an argument to obtain (or rather infer) Born’s rule, based on a simple set of axioms named “Contexts, Systems and Modalities" (CSM). In this approach, there is no “emergence”, but the structure of quantum mechanics can be attributed to an interplay between the quantized number of modalities that is accessible to a quantum system and the continuum of contexts that are required to define these modalities. The strong link of this derivation with Gleason’s theorem was emphasized, with the argument that CSM provides a physical justification for Gleason’s hypotheses. Here, we extend this result by showing that an essential one among these hypotheses—the need of unitary transforms to relate different contexts—can be removed and is better seen as a necessary consequence of Uhlhorn’s theorem.

## 1. Introduction

Many recent articles have proposed derivations of Born’s rule [1,2,3,4], which is clearly a major theoretical basis of quantum mechanics (QMs). In the framework of this Special Issue, let us note, in particular, the construction based on Quantum Darwinism and envariance, as proposed by Wojciech Zurek [5,6,7,8]. It will be discussed further in the conclusion, but in this article we take a different position, i.e., we start from some simple physical requirements or postulates [1], based on established (quantum) empirical evidence [9,10,11,12,13,14,15]; then, we infer a mathematical structure that can describe these physical requirements; and, finally, we deductively obtain Born’s rule and, more generally, the probabilistic structure of QM. We note that related ideas have been discussed in the framework of quantum logic [16,17], but our approach here is a physicist’s rather than a logician’s. With respect to [1], the main purpose of the present article is to simplify further the required mathematical hypotheses, by showing that an essential one—the need of unitary transforms to relate different contexts—can be removed and is better seen as a necessary consequence of Uhlhorn’s theorem, to be introduced below.

## 2. The CSM Framework

The approach of “Contexts, Systems and Modalities” (CSM) is a point of view on Quantum Mechanics based on a non-classical ontology, where physical properties are attributed to physical objects consisting of a system within a context, that is an idealized measurement apparatus. Such physical properties are called modalities, and a modality belongs to a specified system within a specified context, which is described classically (see Annex for more precise definitions). Loosely speaking, the mathematical description of a modality includes both a usual state vector |ψ〉 and a complete set of commuting operators admitting this vector as an eigenstate. Though it may appear heavier at first sight, this point of view eliminates a lot of troubles about QM, and can be seen (in some sense) as a reconciliation between Bohr and Einstein in their famous 1935 debate [18].

The main feature which makes modalities non-classical is that they are both quantized and contextual, as written above. More precisely, the empirical facts that we want to describe mathematically are:(i)in each context a measurement provides one modality among *N* possible ones, that are mutually exclusive. No measurement can provide more than *N* mutually exclusive modalities, and once obtained in a given context, a modality corresponds to a certain and repeatable result, as long as one remains in this same context.(ii)the certainty and repeatability of a modality can be transferred between contexts; this fundamental property is called extracontextuality of modalities. All the modalities that are related together with certainty, either in the same or in different contexts, constitute an equivalence class that we call an extravalence class.(iii)the different contexts relevant for a given quantum system are related between themselves by transformations *g* that have the structure of a continuous group G.

An essential consequence of statement (i), spelled out as Theorems 1 and 2 in [1], is that a probabilistic description is necessary. The main idea is simple: since there are, at most, *N* mutually exclusive modalities in any given context, as well as a continuous infinity of different contexts all carrying *N* modalities, the only way to relate *N* modality in a context to *N* modalities in another one must be probabilistic; otherwise, there would be a “supercontext” with more that *N* mutually exclusive modalities. See [1] for details. This idea is very fundamental in CSM, and it makes that probabilities are a necessary consequence of contextual quantization. From this conclusion, together with statement (ii), we can look for a probability law by using Gleason’s theorem.

The third statement (iii) tells that all the different contexts relevant for a given quantum system are related between themselves by continuous transformations *g*, which are associative, have a neutral element (no change), and have an inverse. Therefore, this set has the structure of a continuous group G, which is generally not commutative (such as the rotations of a macroscopic device). Our goal is then to identify a (non-classical) probabilistic framework [1] corresponding to these requirements, and to draw consequences by using suitable standard theorems.

For this purpose, the central mathematical ingredient is to associate a rank-one projector $P_{i}$ (a N×N hermitian matrix, such as $P^{2}=P=P^{†}$) to each modality, with the rule that modalities associated with orthogonal projectors are mutually exclusive and modalities associated with the same projector are mutually certain. Correspondingly, a context is associated with a set of mutually orthogonal projectors, whereas an extravalence class of modalities is associated with a single projector. In addition, we assume that, given a modality in a context, the probability to get another modality in another context is a function of the two projectors associated with these two modalities (or equivalently with their two extravalence class).

The heuristic motivation for using a complete set of mutually orthogonal projectors to build up a context is that this ensures that the events associated with modalities cannot be subdivided in more elementary events, as this would be the case with classical (partition-based) probabilities. On the other hand, the construction warrants that certainty can be transferred between contexts for extravalent modalities.

Now, we want to show that the usual structure of QM follows from the above hypotheses; this means that unitary transforms between projectors as well as Born’s rule are necessary in the above framework. Let us emphasize that it is easy to show these results fulfill our hypotheses; however, showing that they are necessary requires powerful (and difficult to demonstrate) mathematical theorems. Necessity also means that if one wants to give up unitary transforms or Born’s rule, one has to give up one of the statements above, without contradicting empirical evidence, which is an interesting challenge [19].

## 3. Necessity of Unitary Transforms

As said above, the basic mathematical tool we use is to associate *N* mutually orthogonal projectors with the *N* mutually exclusive modalities within a given context. The choice of such a specific orthogonal set of projectors associated with a context is not given a priori, but once it is done, the sets of projectors in all other contexts should be obtained by a bijective map Γ reflecting the structure of the continuous group G of context changes. For consistency, if two orthogonal projectors are associated with two mutually exclusive modalities, they should stay orthogonal under the map Γ, whatever choice is made for the projectors associated with a “reference" (fiduciary) context. Then, let us consider the following.

**Theorem** **1**(Uhlhorn’s theorem [20,21]). *Let H be a complex Hilbert space with dim(H)≥3, and let $P_{1}\left(H\right)$ denote the set of all rank-one projections on H. Then, every bijective map
Γ: $P_{1}\left(H\right)\to P_{1}\left(H\right)$, such that pq=0 in $P_{1}\left(H\right)$ if and only if Γ(p)Γ(q)=0, is induced by a unitary or anti-unitary operator on the underlying Hilbert space.*

This theorem implies that if orthogonality is conserved as required above, then the transformations between the sets of projectors associated with different contexts is unitary or anti-unitary. (As a reminder, an anti-unitary operator *U* is a bijective antilinear map, such that $\langle Ux|Uy\rangle ={\langle x|y\rangle }^{*}$ for all vectors *x*, *y* in H). In the case of a continuous group of transformations, which is the case here, then the transformation must be unitary (and not anti-unitary) as long as it is continuously connected to the identity, which is the situation we are interested in (see also below).

The strength and importance of Uhlhorn’s theorem is that it requires that the map keeps the orthogonality of rank-one projections, or equivalently of non-normalized vectors (or rays). A transformation mapping an orthonormal basis onto an orthonormal basis is clearly a unitary or anti-unitary transform; however, this result is far from obvious if the conservation of the norm is not required. A related (but weaker) result is Wigner’s theorem, reaching the same conclusion as Uhlhorn’s if the modulus of the scalar product of any two vectors is conserved by the transformation. Uhlhorn’s theorem is much more powerful, since it only assumes that the scalar product is conserved when it is zero, i.e., when the two rays are orthogonal [22].

We thus get a major result: once a set of mutually orthogonal projectors associated with a fiduciary context has been chosen, the sets of projectors associated to all other contexts are obtained by unitary transformations, so we are unitarily “moving” in a Hilbert space. There are also various arguments for using unitary (complex) rather than orthogonal (real) matrices. In our framework, the simplest argument is to require that all permutations of modalities within a context are continuously connected to the identity. This is not possible with (real) orthogonal matrices, which split into two subsets with determinants ±1, but is possible with unitary ones [12,14].

## 4. Necessity of Born’s Rule

The next step is to consider the probability $f(P_{i})$ to get a modality associated with projector $P_{i}$. By construction, a context is such that $\sum_{i=1}^{i=N}P_{i}=I$ and $\sum_{i=1}^{i=N}f\left(P_{i}\right)=1$ for any complete set $\left\{P_{i}\right\}$. However, these are just the hypothesis of Gleason’s theorem, so there is a density matrix ρ such that $f\left(P_{i}\right)=\mathrm{Trace}\left(\rho P_{i}\right)$. More precisely:

**Theorem** **2**(Gleason’s Theorem [23,24]). *Let f be a function to the real unit interval from the projection operators on a separable (real or complex) Hilbert space with a dimension at least 3. If one has $\sum_{i}f\left(P_{i}\right)=1$ for any set $\left\{P_{i}\right\}$ of mutually orthogonal rank-one projectors summing to the identity, then there exists a positive-semidefinite self-adjoint operator ρ with unit trace (called a density operator), such that $f\left(P_{i}\right)=\mathrm{Trace}\left(\rho P_{i}\right)$.*

If we start from a known modality as written in Section 2 above, then the probability value 1 is reached and ρ is also a projector $Q_{j}$, so that $f\left(P_{i}\right)=\mathrm{Trace}\left(Q_{j}P_{i}\right)$ which is the usual Born’s rule. As already explained in [1], we considered initial and final modalities, i.e., rank 1 projectors [14], but, more generally, Gleason’s theorem provides the probability law for density operators (convex sums of projectors), interpreted as statistical mixtures. This clarifies the link between Born’s rule and the mathematical structure of density operators [25]. One, thus, can obtain the basic probabilistic framework of QM; this is enough for our purpose here, but more is needed for a full reconstruction. In particular, composite systems and tensor products should be included. See [19] for a preliminary discussion.

In addition, one must explicitly define the relevant physical properties and associated contexts that may go from space–time symmetries (Galileo group, Lorentz group) to qubits registers. Then, the unitary transforms appear as representations of the relevant group of symmetry [26]. In any case, contextual quantization applies and sets the scene where the actual physics takes place.

## 5. Discussion

For the sake of completeness, it is useful to reinforce the fact that some statements have already been presented in [1]. A key feature of the contextual quantification postulate (see Appendix A) is the fixed value N of the maximum number of mutually exclusive modalities, which turns out to be the dimension of Hilbert space. This provides another heuristic reason for using projectors: the projective structure of the probability law guarantees that the maximum number of mutually exclusive modalities cannot be circumvented.

This would not be the case in the usual partition-based probability theory as partitioning all modalities into N subsets for any given context would not prevent subpartitions, corresponding to additional details or hidden variables forbidden by our basic postulate. This is mathematically equivalent to the Bell’s or Kochen–Specker’s (KS) theorems and all their variants, which essentially demonstrate the inadequacy of probabilities based on partitions. This problem disappears when projectors are used, and then, starting from Gleason’s theorem, there is no choice but Born’s rule.

It should also be noted that Bell’s or KS theorems consider discrete sets of contexts, while Gleason’s theorem is based on the interaction between the continuum of contexts and the quantified number of accessible modalities in a given context. This feature is also fully consistent with the ideas of CSM. Therefore, Gleason’s assumptions in our approach have a deep physical content that combines contextual quantification and extracontextuality of modalities. Since these features are required by empirical evidence, the usual QM formalism provides a good answer to a well-posed question.

We note, however, that our approach leads to some differences with the standard (textbook) one; in particular, the usual quantum state vector |ψ〉 is not predictively complete, since it provides a well-defined probability distribution only when “completed” by the specification of a context [27]. A complete description, also including the contexts requires the use of algebraic methods [19].

To conclude, let us come back to some epistemological difference between the approach used here and the one favored by Wojciech Zurek [5,6,7]. In his point of view, the role of mathematics is prescriptive. First, “Let be Ψ”, and then all the rest should follow. On the contrary, in our approach its role is descriptive; there is a physical world out there, and the mathematical langage is our best tool to “speak” about it—but it is a langage, not the Tables of the Law. Additionally, in CSM, there is no “Emergence of the Classical” [7] as both classical and quantum descriptions are needed to make sense of our physical universe, where an object is a system within a context.

More specifically about Born’s rule, in [8], Wojciech Zurek proposes to derive it, and to identify and analyse origins of probability and randomness in physics, based on environment-assisted invariance (envariance), i.e., a quantum symmetry of entangled systems. An interesting remark in this article is “The only known way to recognize effective classicality in a wholly quantum Universe is based on decoherence. But decoherence is ’off limits’ as it employs tools dependent on Born’s rule. On the other hand, when classicality was ’imposed by force’ by Gleason (…), this seemed to work to a degree, although interpretational issues were left largely unaddressed and doubts have rightly persisted”. The goal pursued in the present paper is to address these interpretational issues with precision and hopefully to remove the corresponding doubts.

These subtleties may appear more philosophical than practical, and they do not preclude an agreement on more down-to-the-earth issues, e.g., the management of decoherence for applications to quantum technologies. However, keeping such foundational issues opened and discussed is certainly a compost for new ideas to germinate.

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
