# Peer review of "Revisiting Born’s Rule through Uhlhorn’s and Gleason’s Theorems"

_entropy, 2022, doi:10.3390/e24020199_

Round 1
Reviewer 1 Report
I appreciate the authors' attempt to explain their work, but I believe the fundamental error remains.
The key ingredient that the authors need to employ Gleason's theorem is the existence of a function f mapping projections to the unit interval [0,1] and having the property that f(P_1) + ... + f(P_N) = 1 for *any* complete set {P_1, ..., P_N}. The CSM definitions, and whatever may be derived from them, cannot provide this very special function.
I will attempt to recast the rather vague CSM description in concrete mathematical terms to illustrate why I believe an additional assumption is needed.
Following the authors, let us define a "quantum system" S as an N-dimensional Hilbert space together with some quantum state rho. (Yes, I know the authors do not wish to start off assuming a quantum state, but I believe it is implicit in their introduction of a "quantum system." At any rate, we won't be using rho explicitly.)
A "context" is a complete set of N orthogonal projections C = {P_1, ..., P_N}, and a "modality" is just an element of the set {1, ..., N} indexing one of these projections.
Given S and C there is a function f such that f(P_m) >= 0 for all m and f(P_1) + ... + f(P_N) = 1. (This is not part of the CSM definitions, but it is implicitly assumed later in Theorem 1 of Ref. [1].) Now, given S and some other context C' = {P'_1, ..., P'_N} there is another function f' such that f'(P'_m) >= 0 for all m and f'(P'_1) + ... + f'(P'_N) = 1. Theorem 1, essentially, claims that there exists some context C' such that 0 < f'(P'_m) < 1 for every P'_m in C'. Fair enough, but, whether this is true or not, what is really needed is to show that f = f' for all contexts C and C'. This, however, cannot be deduced from CSM alone but must come as an additional, and very important, assumption. Indeed, this "noncontextuality" assumption, as it is called, is absolutely essential to Gleason's theorem.
Author Response
We thank the referee for his effort on clarifying this point, helping therefore to identify the issue. The referee writes :
A "context" is a complete set of N orthogonal projections C = {P_1, ..., P_N}, and a "modality" is just an element of the set {1, ..., N} indexing one of these projections.
No, a modality is not ‘just an element of the set {1, ..., N}’, this would be a usual quantum state ‘psi’. A modality is an element of the set, *plus* the specification of the relevant context, as written in the text. Then we define an equivalence class of these modalities, corresponding to being connected with certainty; this equivalence relation is called extravalence. Finally we attribute orthogonal projectors to mutually exclusive modalities (within a context), and the same projector to mutually certain modalities (within an extravalence class).
Given all this, when changing context we look for a probability law f(P_i) which depends only on P_i, which means physically that all modalities in the same extravalence class have the same probability. We do agree that this assumption is absolutely essential to Gleason's theorem, and that it is usually called ‘noncontextuality of probability assignment’. But extracontextuality allows us to introduce this feature in a much more natural way, which is simply that the probability depends on the modality’s extravalence class (and associated projector), and not on the modality’s context.
In our opinion this makes Gleason’s hypotheses much more intuitive and physical, by replacing the strange combination of contextuality for value assignments, and noncontextuality for probability assignments, by the idea of attributing orthogonal projectors to mutually exclusive modalities (within a context), and the same projector to mutually certain modalities (within an extravalence class).
Finally, we emphasize again that in our approach Gleason’s hypotheses are not ‘proven’ (ie, obtained by a deductive reasoning), but ‘justified’, (ie, obtained by an inductive reasoning). Only when these hypotheses are made one can go for a deductive reasoning, get unitary transforms from Uhlhorn and Born’s rule from Gleason, and ultimately check the agreement with experiments.
For clarity some sentences in the section ‘For this purpose…’ on page 2 have been written in bold, and some wordings have been modified.
Reviewer 2 Report
I appreciate that the authors carefully addressed my comments/suggestions.
The current form of the manuscript does not provide a substantial revision –- though I was not expecting one either. Hence, my main criticism remains that a large body of the manuscript repeated previously published work of the authors. I take a neutral stand regarding this point and would be totally fine if the editors decide to accept/reject the manuscript.
A personal remark: as someone who works on quantum probabilities, I do find Uhlhorn’s theorem very interesting, which was largely overlooked by the community. This is certainly a positive input that is worth publishing.
Author Response
We thank the referee for his/her overall positive assessment, and we do agree on the importance of Uhlhorn’s theorem. Making it well known by the quantum foundations community is also an objective of our short paper.
Reviewer 3 Report
Some small comments on the paper entitled – ‘Revisiting Born’s rule through Uhlhorn’s and Gleason’s theorems’.
- This referee understands why i) indeed leads to a probabilistic interpretation. What would we a ‘super context’ though?
- Can the authors maybe spell out a little more what is meant with contextual quantization (also in reference to Postulate 2)?
- May I ask a naïve question: how is the approach of D’Ariano et al. comparing with yours (i.e. the derivation of quantum theory from a operational principles)? Is there a comparison possible – or is this a totally irrelevant question? (Ref: Chiribella, G., D’Ariano, G. M., & Perinotti, P. (2011). Informational derivation of quantum theory. Physical Review A, 84, 012311-1-39)
- When you mention tensor products - for a fuller reconstruction of QM – you could mean – f.i. A tensor product of k isomorphic versions of a Hilbert space H?
Please correct small typo: langage – line 176.
Author Response
We thank the referee for his/her overall positive assessment, here are some answers to his/her questions :
1. We write ‘there would be a ''supercontext'' with more that N mutually exclusive modalities’, and the point is that there is no such supercontext. If there were one, it would be something like a POVM with more than N *mutually exclusive* results, impossible if the dimension is N.
2. Beyond Postulate 2, contextual quantization is a shortcut for the CSM hypotheses, where *both* contextuality and quantization of modalities are required. This is briefly stated in the Appendix, and explained in detail in other CSM papers, that we quote in order to avoid too much repetition.
3. This is certainly not an irrelevant question, but a decent answer would require work going far beyond the scope of this paper; let’s put it in the todo list. In case the referee would be interested, we did a comparison between an old version of CSM (called contextual objectivity) and Hardy’s axioms, see Appendix in https://arxiv.org/abs/quant-ph/0111154 (2001). However these are really old versions and the arguments may be partly obsolete on both sides.
4. Honestly the full treatment of tensor products within CSM is not completed yet, this is also in the todo list. Also including the context in the formalism leads to operator algebras, ‘infinite’ tensor products, and superselection rules, see e.g. von Neumann’s article http://www.numdam.org/item/CM_1939__6__1_0.pdf (especially the introduction), which seems known by mathematicians but apparently not by physicists.
Reviewer 4 Report
The present article makes use of Ulhorn's theorem to show that the CSM approach to axiomatising quantum mechanics can be simplified by removing the requirement of unitarity. This follows from the fact that CSM begins with the association of projectors with modalities, as well as the assumption that mutually orthogonal projectors correspond to mutually exclusive modalities. Ulhorn's theorem states that the only bijections between rank one projectors which preserve orthogonality are either unitary or anti-unitary. This shows that (if one can rule out the anti-unitary case as the authors do) one need not postulate that transformations are unitary if one already assumes the structure of projectors and their orthogonality relations.
Overall I did not find much new content in this article. In the framework of quantum logic it is already know that one can recover essentially all of quantum mechanics just by starting from the fact that propositions (modalitites in CSM) are projectors and that a complete set of exclusive propositions is an orthonormal basis. Then, as the authors state in the paper, Gleason's theorem and Ulhorn theorem allow one to recover the Born rule and unitary transformations.
The only novel aspect of this paper to my mind is that it applies this known fact from quantum logic to CSM.
I do think there is value in highlighting results such as Ulhorn's theorem to a wider readership, but I believe this would be better achieved in a review article which covered the approach of quantum logic in more detail, rather than just applying Ulhorn to CSM.
Author Response
We thank the referee for his/her (mildly) positive assessment. We do agree that similar arguments have been developped in the framework of quantum logic, except maybe Ulhorn's theorem which does not seem very well known. So let’s say that our approach is making these ideas accessible to a physicist’s eye, which is (as far as we can see…) quite blind to quantum logic. Maybe it will make that experimental physicists finally understand quantum logic ? We are not sure yet, but for completeness we added footnote 1 with recent references to quantum logic.
We accept the argument of writing a review article, but clearly this is not suitable for a short paper in this special issue. So this article is rather one more little brick in our CSM construction, to be added later on with others in a review article.
Round 2
Reviewer 1 Report
I greatly appreciate the authors' patience. Unfortunately, I do not see that anything has materially changed in this latest revision.
I maintain that the CSM axioms are insufficient to deduce the Born rule. The authors, in their most recent response, say that they are not attempting to obtain Gleason's hypotheses by deductive reasoning, yet the paper opens by claiming to "get deductively Born's rule." I do not believe they have done this, nor do I believe they can under the CSM axioms.
This manuscript is a resubmission of an earlier submission. The following is a list of the peer review reports and author responses from that submission.
Round 1
Reviewer 1 Report
As a friend of Wojciech Zurek, I would like to thank the authors, Dr. Auffeves and Dr. Grangier, for contributing the article to this special issue.
The authors discussed a derivation of Born’s rule within the “context, systems and modalities” framework developed in their previous work [1]. The majority of the current article contains summaries/discussions of the published results in [1], with new input to weaken one of their previous assumptions of “unitary transforms” by using Uhlhorn’s theorem.
In this context typically Wigner’s theorem is involved. It is interesting to see that the conditions of Wigner can be further relaxed by Uhlhorn -- This is certainly of value to the community of motivating/deriving quantum mechanics from fundamental principles. However, compared to the published work [1], I am not entirely sure the new material is adequate to warrant publication as a research article.
Of course, the discussion is in context with the scope of the special issue, and the editors may find it interesting and suitable for acceptance.
My other little concern is the discussion of Quantum Darwinism (QD) and the result of the present paper. These include the sentences in Line 13 and the last paragraph. In my view, QD deals with the emergence of the classical reality given quantum mechanics in the first place, rather than deriving quantum mechanics (or particularly Born’s rule). Hence I find it confusing to speak of constructing Born’s rule based on QD, though the cited references [6-8] on QD did mention Born’s rule. -- This point could be clarified in the aforementioned places.
Nevertheless, motivating Born’s rule is a topic Zurek is interested in and he has contributed to this field as well. For instance, [PHYSICAL REVIEW A 71, 052105], where he derived Born’s rule using entanglement structures. This may be of more relevance to the present manuscript.
A minor point: Line 100, "... is clearly a unitary transform." -- It can be anti-unitary as well.
Reviewer 2 Report
The authors provide an extension of their previous work in which they claim to "get deductively Born's rule" from their CSM axioms. The present work claims to extend this by using Uhlhorn's theorem in place of the assumption that all contexts, represented as complete orthonormal bases, are related via unitary transformations. The axioms are stated in the appendix.
Although the focus of the paper is on the use of Uhlhorn's theorem, the real issue with the paper is the authors' use of Gleason's theorem to derive the Born rule. In truth, this is an issue with their prior work, Ref. [1], but the mistake is repeated here and is central to their argument.
The first paragraph of Sec. 4 states that "The next step is to consider the probability f(P_i) to get a modality associated with
projector Pi. By construction a context is such that sum_{i=1}^N P_i = I, and sum_{i=1}^N f(P_i) = 1 for any complete set {P_i}." From this, they invoke Gleason's theorem to deduce the Born rule.
The property sum_{i=1}^N P_i = I does indeed follow from the stated mathematical representation of a context. However, the property sum_{i=1}^N f(P_i) = 1 is not derivable from the given axioms. Indeed, the axioms say nothing of probability. So, while one may define a frame function f() that has the required properties and say that f(P_i) is the probability of obtaining the modality P_i, this amounts to including an additional axiom. This additional axiom, by virtue of Gleason's theorem, amounts to assuming the Born rule, so its "derivation" becomes a circular argument.
Gleason's original argument was to assume a probability function (what he calls a "measure"), define a frame function in terms of this "measure", and then deduce its form from the fact that non-negative frame functions must be regular. The authors have no such starting point.
Because of this fundamental error, I cannot recommend the paper for publication.
Reviewer 3 Report
This short, well-written article addresses the Context, Systems, and Modalities (CSM) conceptual approach to Quantum Mechanics (QM). This framework has been introduced and discussed in some previous publications, the contribution of the submitted manuscript being to present Uhlhorn's theorem as the reason why unitary transformations are ubiquitous/mandatory in QM. This theorem allows for the requirement that “if two orthogonal projectors are associated with two mutually exclusive modalities, they should stay orthogonal under the map G” to replace the requirement of unitary transformations as an ad hoc rule. This is indeed a conceptual advance in the CSM framework.
Although I am not sure about the effectiveness of this alternative approach to produce new physics, its certainly important to look for more “natural” postulates for QM. In this spirit, I recommend the publication of the submitted manuscript.